# The Clinical Value of Multimodal Ultrasound for the Differential Diagnosis of Hepatocellular Carcinoma from Other Liver Tumors in Relation to Histopathology

**DOI:** 10.3390/diagnostics13203288

**Published:** 2023-10-23

**Authors:** Marinela-Cristiana Urhuț, Larisa Daniela Săndulescu, Adriana Ciocâlteu, Sergiu Marian Cazacu, Suzana Dănoiu

**Affiliations:** 1Department of Gastroenterology, Emergency County Hospital of Craiova, Doctoral School, University of Medicine and Pharmacy of Craiova, 200349 Craiova, Romania; cristiana.urhut@yahoo.com; 2Department of Gastroenterology, Research Center of Gastroenterology and Hepatology, University of Medicine and Pharmacy of Craiova, 200349 Craiova, Romania; adriana_ciocalteu@yahoo.com (A.C.); cazacu2sergiu@yahoo.com (S.M.C.); 3Department of Pathophysiology, University of Medicine and Pharmacy of Craiova, 200349 Craiova, Romania; suzanadanoiu@yahoo.com

**Keywords:** multimodal ultrasound, contrast-enhanced ultrasound, elastography, liver tumors, hepatocellular carcinoma

## Abstract

Recent advances in the field of ultrasonography offer promising tools for the evaluation of liver tumors. We aim to assess the value of multimodal ultrasound in differentiating hepatocellular carcinomas (HCCs) from other liver lesions. We prospectively included 66 patients with 72 liver tumors. The histological analysis was the reference standard for the diagnosis of malignant liver lesions, and partially for benign tumors. All liver lesions were assessed by multiparametric ultrasound: standard ultrasound, contrast-enhanced ultrasound (CEUS), the point shear wave elastography (pSWE) using shear wave measurement (SWM) method and real-time tissue elastography (RTE). To diagnose HCCs, CEUS achieved a sensitivity, specificity, accuracy and positive predictive value (PPV) of 69.05%, 92.86%, 78.57% and 93.55%, respectively. The mean shear-wave velocity (Vs) value in HCCs was 1.59 ± 0.29 m/s, which was lower than non-HCC malignancies (*p* < 0.05). Using a cut-off value of 1.58 m/s, SWM achieved a sensitivity of 54.76%, and 82.35% specificity, for differentiating HCCs from other malignant lesions. The combination of SWM and CEUS showed higher sensitivity (79.55%) compared with each technique alone, while maintaining a high specificity (89.29%). In RTE, most HCCs (61.53%) had a mosaic pattern with dominant blue areas corresponding to type “c” elasticity. Elasticity type “c” was 70.59% predictive for HCCs. In conclusion, combining B-mode ultrasound, CEUS, pSWE and RTE can provide complementary diagnostic information and potentially decrease the requirements for other imaging modalities.

## 1. Introduction

Over time, ultrasound (US) has shown tremendous development. Currently, US allows not only the assessment of liver anatomy and morphology, but also the characterization of tissue elastic properties and the perfusion of tumors through elastography and contrast-enhanced ultrasound [1].

With these innovative methods added to greyscale ultrasound, multiparametric ultrasound has the potential to serve as a “one-stop shop” algorithm for the diagnosis of focal liver lesions (FLLs) [1].

In the field of ultrasonography, contrast-enhanced ultrasound represents a major breakthrough. CEUS provides real-time evaluation of FLLs at a higher spatial resolution when compared with computed tomography (CT) or magnetic resonance imaging (MRI) [2,3]. Additional advantages include the safety profile and the lack of nephrotoxicity of the contrast agent [3]. The value of CEUS in differentiating malignant and benign tumors has been well established [4,5]. A meta-analysis of 57 studies published in 2018 reported that the pooled sensitivity and specificity of CEUS for the characterization of liver tumors was 92% and 87% [4]. The diagnostic performance of CEUS is comparable with that of contrast-enhanced computed tomography (CECT) and contrast-enhanced magnetic resonance imaging (CEMRI) [5]. Recent evidence indicates that CEUS is a valuable tool in diagnosing early hepatocellular carcinomas, with a sensitivity of 92%, a specificity of 93% and an AUC (area under the curve) of 0.95 [6].

Elastographic techniques based on ultrasound have been extensively described in liver stiffness assessment by the World Federation for Ultrasound in Medicine and Biology (WFUMB), as well as by the European Federation of Societies for Ultrasound in Medicine and Biology (EFSUMB), whose current guidelines provide recommendations on the use of elastography in clinical medical practice [7,8]. Point shear-wave elastography (pSWE) is an ultrasound-based method that uses acoustic radiation force impulses to create shear waves [7]. Several meta-analyses were conducted to investigate the clinical utility of shear wave elastography in FLL characterization, with promising results regarding the discrimination of malignant and benign lesions [9,10,11]. However, the role of pSWE in the assessment of liver tumors is still controversial due to the overlap of stiffness values; therefore, international guidelines cannot provide guidance in this regard [8].

The SWM method is a pSWE technique introduced in Hitachi Arrietta (Hitachi Aloka Medical Ltd., Tokyo, Japan) [8]. The distinctive feature of this method is the VsN, a reliability index that indicates the percentage of effective shear wave velocity. The VsN should have values ≥ 50% to consider the measurement reliable [12]. Although there are studies regarding the application of SWM in liver fibrosis [13] and thyroid tumors [14], we did not find any study that evaluates the utility of SWM in FLLs.

Real-time tissue elastography is a technique based on ultrasound that enables a visual evaluation of tissue elasticity and can provide novel information when added to standard ultrasound examination [15]. This method showed promising results in the diagnosis of prostate cancer [16], thyroid nodules [17] and staging of liver fibrosis [18], as confirmed in meta-analyses of published papers. However, studies regarding the role of RTE in the differential diagnosis of liver tumors are limited [19,20,21,22,23], and some are conducted in the intraoperative setting [21,22,23].

Although there is an increasing trend in using multiparametric ultrasound in liver tumors, there are still little data available. In this study, we aimed to evaluate the value of multimodal ultrasound, based on standard ultrasound, CEUS, SWM and RTE in differentiating hepatocellular carcinomas from other liver lesions.

## 2. Materials and Methods

This single-institution prospective study included patients with focal liver lesions, evaluated between November 2022 and June 2023, in a tertiary gastroenterology and hepatology department. Of 102 initially considered liver lesions, 72 were enrolled in our analysis (Figure 1). The study was approved by the Ethical Committee of the University of Medicine and Pharmacy of Craiova.

The inclusion criteria comprised focal liver lesions of ≥ 2 cm, visible on standard ultrasound, located within 8 cm from the transducer, and histopathological confirmation of the diagnosis for malignant tumors.

We excluded from the study patients with impaired clinical status, cases with uncertain diagnosis at the time the study was completed, previously treated liver tumors and infiltrative lesions with reduced visibility in standard ultrasound. The other exclusion criteria were outlined based on the limitations of sonoelastographic techniques: dyspnea, a patient’s inability to hold their breath, the size of an FLL being ≤ 2 cm (as the region of interest (ROI) in point shear wave elastography is 1 × 1.5 cm) and the depth of the lesions being more than 8 cm from the skin surface.

### 2.1. Multimodal Ultrasound Examination

Ultrasound investigations were performed using a Hitachi Arietta V70 system (Hitachi Ltd., Tokyo, Japan) equipped with the convex probe C251. Ultrasound examinations were performed by an experienced sonographer (level III, according to EFSUMB classification).

The imaging protocol started with examining the entire liver in greyscale ultrasound to identify the target lesion. The following tumor characteristics were assessed: size, echogenicity, echotexture, boundary, presence of a halo sign, satellite nodules and ancillary features favoring HCC such as “nodule-in-nodule” and mosaic appearance.

For point shear-wave and real-time tissue elastography, the patient was placed in the supine position or in slight lateral decubitus, with the right arm in extension above the head. The ultrasound transducer was positioned in a right intercostal space. However, the probe’s position was adjusted according to the tumor location in certain situations. An epigastric approach was used for liver lesions in the left liver lobe. After visualization of the liver parenchyma and tumor detection in standard ultrasound, SWM mode was activated from the ultrasound menu, to perform point shear wave elastography. Areas of tumoral necrosis, artefacts, biliary ducts or small vessels were avoided. Ten shear wave measurements were performed both in the tumor and liver parenchyma, during expiratory apnea. The final results were reported as the median shear wave velocity (Vs), displayed as m/s, together with the measurement efficacy rate (VsN), expressed as a percentage.

For elasticity imaging with real-time tissue elastography, Elasto mode was activated. The screen was split into a B-mode US image on the left and an elasticity image on the right. The region of interest was placed at the level of the target lesion. Light pressure was applied to generate a color map superimposed on the tumour, in which red encoded soft tissue, areas with moderate strain were shown as green and hard tissue was displayed in blue. Elasticity images that showed the most stability during the examination and corresponded to the downward slope of the strain graph were selected. We did not focus on the difference in elasticity between the liver parenchyma and the liver lesion. The color patterns of the liver lesions were classified based on the findings from two classification systems previously reported in the literature [22,23].

The elasticity type of liver tumor (ETLT) classification, developed by Kato K. et al. [23], distinguishes four elasticity types of liver tumors: type A, characterized by a homogenous green pattern; type B, corresponding to lesions with a mosaic pattern with dominant green areas; type C, suggestive of a mosaic pattern with predominantly blue areas; type D, allocated for lesions that appear homogenously blue. Omichi K. et al. created the modified elasticity type of liver tumor (modETLT) classification, consisting of six types, some of which correspond to those reported by Kato K. He further described a homogenous light blue pattern, that we also identified in the elasticity images in our study [22].

After we evaluated the color patterns of the liver lesions in our study, we categorized them into five types, detailed in Table 1. Type “a”, “b”, “c”, and “e” are similar to Kato’s ETLT; meanwhile, type “d” corresponds to type 5 of the modETLT classification. The elasticity types in our study are illustrated in Figure 2, Figure 3, Figure 4, Figure 5 and Figure 6. The elasticity images were classified by two investigators, one of whom was blinded to the clinical data. Discordant results were reviewed by a third reader, who made the final decision.

Contrast-enhanced ultrasound examinations and the interpretation of CEUS results was performed according to the Guidelines and Good Clinical Practice Recommendations for CEUS in the Liver, updated in 2020 [24]. The contrast agent used was SonoVue (Bracco, Geneva, Switzerland) at a dose of 1.6 mL.

### 2.2. Reference Standard Method

The reference standard method for malignant liver lesions was the histological analysis. Histopathological assessment was necessary for hepatocellular carcinomas due to inconclusive imaging criteria or to guide oncological therapy in patients who were not eligible for surgical treatment or locoregional therapies.

For liver metastases, the indications for histopathological evaluation were as follows: unknown primary tumor site after detailed investigations; primary tumor not accessible for biopsy; history of extrahepatic malignancy and indeterminate liver lesions on imaging tests.

One atypical focal nodular hyperplasia (FNH) and two liver adenomas with inconclusive imaging findings were also diagnosed by histopathology. For the other benign liver tumors, the diagnosis was determined by CT or MRI.

### 2.3. Statistical Analysis

For data analysis, the IBM program- Statistical Analysis Software Package (SPSS), version 29.0 for Windows, was used together with Microsoft Excel 2019 (Microsoft Office Professional Plus 2019). The sensitivity, specificity, accuracy, PPV and negative predictive value (NPV) of CEUS, SWM and the combination of CEUS and SWM were calculated. A one-way ANOVA test was employed to compare age between patient groups. The Mann–Whitney U test was conducted to assess the difference in age between the gender groups and to compare the differences in Vs, depth and size of the lesions in the two groups formed, according to VsN value (<50, ≥50). The Kruskal–Wallis test was run to evaluate the differences in SWM values between HCCs, non-HCC malignancies and benign tumors, as our data were not normally distributed. A paired-samples *t*-test was conducted to determine whether there was a significant mean difference between HCCs and liver parenchyma. The Fisher’s exact test assessed the association between the presence of liver cirrhosis and VsN. The results were considered statistically significant if *p* was < 0.05. Testing for the normality of data was undertaken using the Shapiro–Wilk test (*p* > 0.05).

## 3. Results

Sixty-six eligible patients with seventy-two liver tumors were enrolled in the study. Among the 66 patients, 56 had liver tumors with a high suspicion of malignancy but inconclusive imaging tests or had been referred to our institution for percutaneous liver biopsy. The other ten were consecutive patients, with liver lesions diagnosed as benign on imaging studies. One patient presented with both malignant and benign tumors. The distribution of liver tumors according to the final diagnosis is illustrated in Table 2.

Most patients were male (67.70%) with a mean age of 66.36 ± 11.77 years. No significant difference in age was found between genders or the three groups of liver tumors (*p* > 0.05).

Liver cirrhosis was present in 29 patients (43.93%). In a subset of patients (9.68%) who were not clinically diagnosed with any underlying liver disease, histological analysis revealed pathological changes compatible with chronic hepatitis. Ascites and portal vein thrombosis were found in 14 and 12 patients, respectively. Ascites was present in small amounts that did not contraindicate percutaneous biopsy of the hepatic tumor. A history of extrahepatic malignancies was noted in 15 patients. The characteristics of the patients are detailed in Table 3.

### 3.1. B-Mode Ultrasound

In the majority of cases (90.9%), a single lesion was analyzed in standard ultrasound, while in six patients, two lesions were evaluated. The mean size of the lesions was 70.11 ± 38.97 mm. No statistically significant difference in the size of lesions was observed between HCCs, non-HCC malignancies and benign tumors (*p* = 0.183). Ancillary features favoring HCCs were assessed. “Nodule-in-nodule” architecture was observed in 18 lesions (25%), from which 17 cases were HCCs, with only one being an intrahepatic cholangiocarcinoma (iCCA). A mosaic appearance was observed in 16 lesions, all of which were HCC cases (Figure 7). The ultrasound features of liver tumors are illustrated in Table 4.

### 3.2. Contrast-Enhanced Ultrasound

CEUS correctly diagnosed eight cases of benign liver tumors (Figure 8). Two liver adenomas, which developed in patients at risk for HCCs (chronic hepatitis B and liver cirrhosis, respectively), displayed arterial hyperenhancement and mild washout in the late phase and were wrongly classified as hepatocellular carcinomas. One focal nodular hyperplasia was misdiagnosed as a malignancy.

The hallmarks of HCCs are represented by diffuse arterial hyperenhancement, with chaotic intratumoral vascularity and a late onset (>60 s), mild washout [24]; these were detected in 29 out of 44 hepatocellular carcinomas (65.9%), from which 17 nodules (38.63%) were developed in a cirrhotic liver and 12 in patients without liver cirrhosis (27.27%) (Figure 9).

For two lesions, interpretation was impossible due to image degradation, and these cases could not be categorized.

In 11 nodules (25%), the CEUS pattern suggested malignancy, but was not characteristic of HCC. Two cases of HCC associated with liver cirrhosis presented arterial hyperenhancement but no washout. (Figure 9).

Considering the typical pattern of HCCs, CEUS achieved a sensitivity of 69.05%, a specificity of 92.86% and 78.57% accuracy (Table 5).

All ten liver metastases were correctly diagnosed by contrast-enhanced ultrasound. Three arterial phase enhancement patterns were identified: diffuse heterogeneous hyperenhancement in hypervascular metastases (five lesions), rim-like hyperenhancement and hypoenhancement in hypovascular metastases. Early (<60 s) and marked washout was noted in all cases. Figure 10 illustrates a representative example of multiparametric ultrasound evaluation of liver metastases.

Intrahepatic cholangiocarcinoma exhibited rim arterial hyperenhancement in two cases, diffuse heterogeneous hyperenhancement in three cases, and one showed diffuse heterogeneous hypoenhancement. Early washout was present in five lesions. However, in one iCCA, the onset of washout was late (>60 s) but with a marked degree.

Hepatic lymphoma showed inhomogeneous hyperenhancement, followed by early and marked washout.

### 3.3. Shear Wave Measurements in Liver Tumors

Liver lesions had a wide range of SWM values, as shown in Table 6, with overlapping results between benign and malignant tumors. The Kruskal–Wallis H test, followed by a post hoc test for pairwise comparison, showed no statistically significant difference between benign tumors and HCCs (*p* = 0.3) or benign tumors and non-HCC malignancies (*p* = 0.28). However, the SWM values were higher in the non-HCC malignancy group, compared to the hepatocellular carcinomas (*p* = 0.03) (Figure 11).

The results indicated higher SWM values in the cirrhotic liver (2.12 ± 0.45 m/s) compared with the HCC nodules (1.54 ± 0.24 m/s), with a significant statistical difference between them (*p* < 0.001). In non-cirrhotic patients, there was no significant difference between Vs in HCCs (1.88 ± 0.41 m/s) and the background liver (1.73 ± 0.41 m/s, *p* = 0.33).

A receiver operating characteristic (ROC) curve was applied to determine the optimal cut-off value of Vs, in order to differentiate between HCCs and non-HCC malignancies (Figure 12). Using the maximum value of the Youden index (sensitivity + specificity − 1), the optimal threshold was set at 1.58 m/s. The ROC curve analysis showed that the area under the curve (AUC) was 0.70. Point shear wave elastography correctly identified 23 of 44 (52.27%) hepatocellular carcinomas and 14 of 17 (82.35%) non-HCC malignancies. The sensitivity, specificity and accuracy of SWM for the correct characterization of HCCs were 54.76%, 82.35% and 62.71%, respectively (Table 7).

In 51.38% of elastographic measurements performed in liver tumors (37 cases), VsN had values under 50%. The cases were divided into two groups according to VsN: VsN ≥ 50%, VsN < 50%. Higher Vs, higher depth values and lower tumor sizes were associated with a VsN < 50, but the difference was not statistically significant (*p* > 0.05). Also, there was no statistically significant difference between the groups (*p* > 0.05) regarding the presence of liver cirrhosis (Table 8).

### 3.4. Real-Time Tissue Elastography

RTE was considered invalid in eight lesions, in which frames were not stable and were excluded from analysis.

The distribution of liver tumors, according to their elasticity color code, is shown in Table 9.

Benign tumors showed type “a” (three liver hemangiomas and one complicated liver cyst) and “b” (one liver hemangioma and one liver adenoma) elasticity patterns. Only one focal nodular hyperplasia was type “c”.

Most HCCs were classified as type “c” (61.53%). In the non-HCC malignancy group, type “c” and “e” were dominant.

Although the number of benign tumors and non-HCC malignancies is insufficient to perform an analysis, most malignant tumors were classified as type “c” and “e”, indicating that these patterns are suggestive of malignancy.

Type “c” elasticity was 70.59% predictive for HCCs (PPV, 95% CI: 58.30–80.47%), with 61.54% (95% CI: 44.62–76.64%) sensitivity, 60% (95% CI: 36.67–78.87%) specificity, 60.94% (95% CI: 47.93–72.90%) accuracy, and a negative predictive value of 50% (95% CI: 37.52–61.48%).

Interrater agreement on the classification of liver tumors, according to the elasticity type, was evaluated using Cohen’s kappa (κ) and the results showed substantial agreement (κ = 0.791, *p* < 0.001).

### 3.5. The Added Value of the SWM Method to Contrast-Enhanced Ultrasound Assessment of HCC

In SWM combined with CEUS, the threshold Vs value (1.58 m/s) was applied to reclassify cases in which CEUS indicated malignancy, but the pattern of vascularization was not specific for a definite diagnosis (atypical HCCs cases, cholangiocarcinomas and liver lymphoma). Using this diagnostic approach, the number of HCC cases correctly identified increased from 29 (65.90%), when using only CEUS, and 23 (52.27%), when using only SWM, to 35 (79.54%). Furthermore, 16 (94.11%) non-HCC malignancies were correctly identified. The combined use of CEUS and SWM showed increased sensitivity, accuracy and NPV for HCC diagnosis, compared with each technique used alone (Table 10).

## 4. Discussion

B-mode ultrasound is proposed in international society guidelines as the main surveillance tool for patients at risk of developing HCCs [25,26]. Currently available data suggest a limited sensitivity of surveillance US in detecting early-stage hepatocellular carcinoma, with a value of 63% in one study, and 47% according to another meta-analysis [27,28].

Certain auxiliary features can guide the diagnosis even from standard ultrasound. A mosaic appearance and “nodule-in-nodule” architecture are characteristic features of HCCs that can be seen in US, as well as in other imaging techniques [29]. Other ultrasound findings in advanced hepatocellular carcinoma are a lateral shadow, a halo sign as the corresponding sonographic expression of a fibrous capsule and posterior eco enhancement. Still, these patterns are not HCC-specific, as they can also be observed in other liver tumors: hemangiomas also exhibit posterior acoustic enhancement, and the halo sign can be noticed in 40% of liver metastases [30,31,32]. In our study, mosaic appearance was observed only in HCC nodules. “Nodule-in-nodule” appearance was observed in 25% of liver tumors, mostly HCCs (23.61%). Almost half of HCCs (47.72%) presented a halo sign.

EASL guidelines propose CEUS as a second diagnostic modality for HCCs developed in cirrhotic patients after CT and MRI [25]. The key features of HCCs in CEUS are diffuse hyperenhancement, followed by washout with a late onset (>60 s) and mild intensity [25].

There is limited literature data regarding the CEUS findings of HCCs developed in non-cirrhotic livers [33]. Patients usually present with larger nodule sizes, as they are not included in surveillance programs. The enhancement behavior in CEUS is similar to that in HCCs developed in the context of liver cirrhosis [34]. However, a multicenter study by Dong Y et al., including 96 patients, suggested that HCCs, developed in the non-cirrhotic liver, show arterial phase hyperenhancement with rapid washout [35].

In our study, the typical enhancement pattern of HCCs on CEUS was observed in 65.90% of all HCC nodules. The majority of HCCs developed in patients without liver cirrhosis (63.15%) presented with arterial hyperenhancement and late, mild washout. Considering the typical pattern of HCCs, CEUS achieved a sensitivity of 69.05%, a specificity of 92.86% and 78.57% accuracy. In another study, based on 64 focal liver lesions with a histopathologically proven diagnosis comprising 53 HCCs, CEUS showed similar diagnostic performance compared with our results, with a sensitivity, specificity, accuracy and PPV score of 73.6%, 90.9%, 76.6% and 97.5%, respectively [36]. Similar results were found by Vidili G. et al. when assessing category CEUS LR-5 [37], which, according to CEUS Liver Imaging Reporting and Data System (LI-RADS) algorithm released by the American College of Radiology, includes nodules ≥10 mm with a typical enhancement pattern of HCCs [38]. However, that study included a significantly higher number of patients, and liver cirrhosis was present in all of them [37].

Two false positive cases, classified as HCCs by CEUS, were histopathologically diagnosed as liver adenomas. Both lesions were developed by patients at high risk for HCCs: one patient had chronic hepatitis B and the other had liver cirrhosis. A liver adenoma developed in the context of liver cirrhosis is an extremely rare situation [39]. Liver adenoma usually shows arterial hyperenhancement, followed by isoenhancement in the portal venous and late phase. However, a significant number can demonstrate washout in the late phase. In these circumstances, the definitive diagnosis requires histological analysis [24,40].

In our group, 13 hepatocellular carcinomas were not diagnosed by CEUS examinations due to atypical enhancement patterns. In terms of arterial enhancement, four lesions presented peripheral enhancement, and another showed isoenhancement. Regarding washout, two nodules lacked washout, and another ten HCCs presented early and marked washout. Rim-like arterial phase enhancement is not specific to HCCs. Early marked washout may suggest a poorly differentiated HCC or indicate non-HCC malignancy such as liver metastasis or intrahepatic cholangiocarcinoma [24,41,42]. However, liver metastases are rare in a cirrhotic liver, representing 1.7% of tumors [43]. Furthermore, in a non-cirrhotic liver, in the context of extrahepatic malignancy, the diagnosis of liver metastases can be easily established through CEUS [24]. Therefore, in a patient with advanced chronic disease, atypical HCC requires mainly differentiation with cholangiocarcinoma, and liver biopsy is usually necessary to accurately distinguish between these entities [25,44].

Based on the theory that a hepatocellular carcinoma is a soft tumor in contrast to the firm consistency of cirrhotic liver, as well as the hypothesis that liver metastases and cholangiocarcinomas are stiff lesions due to their desmoplastic stromal response, elastography can provide complementary information regarding tumoral stiffness and help in the differential diagnosis [45,46]. Our study found a statistically significant difference in SWM values between HCCs and non-HCC malignant tumors (1.59 ± 0.29 m/s. versus 1.92 ± 0.42 m/s). Higher stiffness values for cholangiocarcinoma and liver metastases, when compared with HCCs, were also reported in another study, although statistically significant differences were found only between hepatocellular carcinomas and colon cancer metastases [47]. Another research demonstrated that a cholangiocarcinoma is the stiffest malignant tumor; however, these results were achieved using a 2D shear wave elastographic method [48].

In previous research, the shear wave velocity value of HCCs ranged from 2.15 m/s ± 0.75 m/s to 3.07 ± 0.89 m/s [46,49]. Our results showed lower values than previously reported. This could be explained by the fact that we predominantly included large hepatocellular carcinomas in which necrosis and hemorrhage are frequent and could influence stiffness values [50]. The Vs was significantly lower in HCC than the background parenchyma in cirrhotic patients, consistent with results from other studies [46,51].

As previously mentioned, liver metastases are generally firm tumors due to desmoplastic stroma. However, some liver metastases show small amount of fibrous stroma [52]. In our group, the lowest Vs value of 1.36 m/s was observed in a patient with liver metastases from a small-cell lung carcinoma. This explanation lies in the fact that a small-cell lung carcinoma is a soft, friable tumor with extensive necrosis [53]. Furthermore, some small cholangiocarcinomas show increased cellularity and less fibrous stroma [54,55]. These peculiarities could explain the variability of Vs in the non-HCC malignancies group and should be considered when interpreting the results of an elastographic examination.

Using a cut-off value of 1.58 m/s, the SWM method achieved a sensitivity, specificity and accuracy of 54.76%, 82.35% and 72.71%, respectively, for differentiating HCCs from other malignant tumors. SWM diagnostic performance was inferior to CEUS.

In the present study, SWM combined with CEUS, in malignant liver lesions with unspecific enhancement patterns, significantly increased the sensitivity for HCC diagnosis, compared with both methods used separately, while maintaining a high specificity. These findings suggest that SWM complements CEUS in the evaluation of liver tumors and may be a feasible tool for clinicians to improve diagnostic performance in particular cases.

Our research showed no significant statistical difference in Vs between benign and malignant tumors, which is possibly related to the various histological characteristics of benign lesions. The four hemangiomas included in our study had a wide range of Vs from 1.54 to 2.51 m/s, with a mean value of 2.02 m/s. Liver hemangiomas can show areas of fibrosis, thrombosis and calcifications, more commonly in large lesions. In some atypical hemangiomas, the amount of fibrosis can be significant and can transform the tumor into a fibrotic nodule, as in the case of sclerosing hemangioma [45,56]. Frulio N. et al. also reported a mean value of acoustic radiation force impulse (ARFI) measurement greater than 2 m/s for hepatic hemangiomas [57]. Focal nodular hyperplasia is a stiff lesion, due to the presence of a central stellate scar and radial fibrous septa. The lesion was compared with a focal area of liver cirrhosis [58]. However, in the two FNHs from our study, with sizes of 30 mm and 27 mm, respectively, Vs was 1.43 m/s and 1.58 m/s, which could be related to the fact that some small FNHs can lack the central scar [59]. Elastography remains the last step in the differentiation of benign tumors, as most cases have a typical appearance in CEUS/CT/MRI.

In the SWM method, the quality of a measurement is evaluated based on the VsN. The result is considered reliable if the VsN is ≥50% [12]. The defined ROI is assumed to be placed in a homogenous area; however, liver tumors have a different architecture than liver parenchyma and may be heterogenous, with areas of necrosis and hemorrhage [57,60]. The SWM elastographic method was studied in thyroid tumors, where the results demonstrated higher Vs and lower VsN values in malignant lesions, compared with benign ones [14]. Another study showed that a VsN < 50% is associated with higher abdominal circumference, increased body mass index (BMI), and a distance between the transducer and liver greater than 2 cm [12]. In 51.38% of cases from our study (*n* = 37), the VsN had values below 50%. No statistically significant differences were found regarding the presence of liver cirrhosis, Vs, the size and the depth of lesions between the low VsN group (<50%) and high VsN group (≥50%).

Kato K. et al. proposed the ETLT classification for the differential diagnosis of liver tumors using real-time tissue elastography in the intraoperative setting. His work showed that RTE is useful in differentiating hepatocellular carcinomas from metastatic adenocarcinomas. The majority of HCCs showed a mosaic pattern with dominant green areas, corresponding to type B in the ETLT classification; meanwhile, metastatic adenocarcinomas were classified either as type C, represented by the mosaic pattern with a dominant blue pattern, or type D, corresponding to homogenous blue. Regarding type B as HCCs, the sensitivity, specificity and accuracy were 95.5%, 90.9% and 92.7%, respectively. Considering type C and D as metastatic adenocarcinomas, the results showed 100% sensitivity, 80.6% specificity and 89.1% accuracy [23].

On the contrary, in the study undertaken by Wang J. et al., most HCCs presented a mosaic pattern with a dominant blue area. Results showed a sensitivity of 87.2%, a specificity of 65.6% and an accuracy of 77.55% [19]. Also, in our study, more than half of the HCCs (51.53%) presented a similar color pattern (type “c” elasticity); however, we achieved lower sensitivity, specificity and accuracy than that achieved by Wang J. et al. This can be explained by the fact that a significant proportion of HCCs (38.46%) presented as other elasticity types (namely, “b” and “e”) and also, the fact that some liver metastases and cholangiocarcinomas had similar color patterns as the HCCs.

This study has limitations. Firstly, this is a monocentric study that included a small number of liver lesions. However, one of the inclusion criteria was the histological diagnosis of malignant tumors, which significantly restrained the number of tumors we could have introduced, as currently benign tumors, hepatocellular carcinomas and liver metastases can be diagnosed non-invasively in specific clinical circumstances. Also, the limitations of pSWE, concerning the size and depth of lesions, further decreased the number of lesions. Another limitation is represented by the imbalance of liver tumors classes, with a significantly lower number of lesions in the benign and non-HCC malignancies group; however, our main focus was on hepatocellular carcinomas. Thirdly, we included both cirrhotic and non-cirrhotic patients, which can be a source of heterogeneity in the study sample.

Despite these limitations, a key strength of our study is that all malignant tumors were confirmed through histological analysis. Furthermore, the study was conducted in a tertiary center and all lesions were analyzed by an experienced sonographer at our institution.

## 5. Conclusions

In conclusion, multimodal ultrasound has the potential to provide a complete assessment of focal liver lesions within a single examination. Atypical liver tumors could benefit the most from multiparametric ultrasound, as the integration of data achieved from standard ultrasound, contrast-enhanced ultrasound, point shear wave elastography and real-time tissue elastography can clarify an otherwise unclear diagnosis. Given the fact that multiparametric ultrasound is a newly introduced concept, multicentric studies are required to evaluate its clinical applications in liver cancer.

## Figures and Tables

**Figure 1 diagnostics-13-03288-f001:**
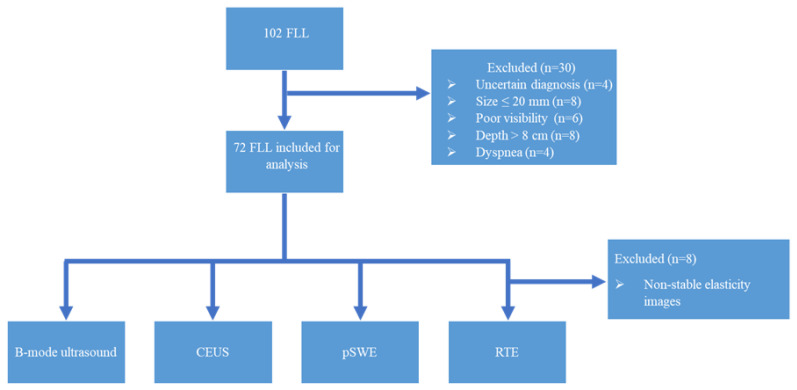
Flowchart of the study protocol. FLL: focal liver lesions; CEUS: contrast-enhanced ultrasound; pSWE: point shear-wave elastography; RTE: real-time tissue elastography.

**Figure 2 diagnostics-13-03288-f002:**
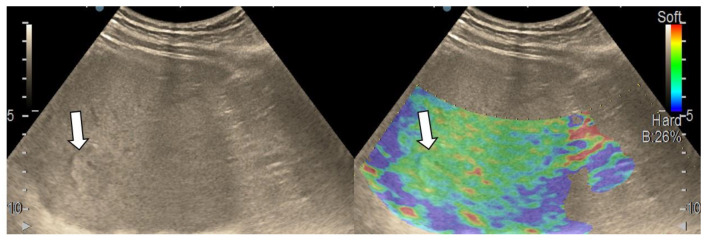
Liver hemangioma with type “a” elasticity (homogenously green) (arrows).

**Figure 3 diagnostics-13-03288-f003:**
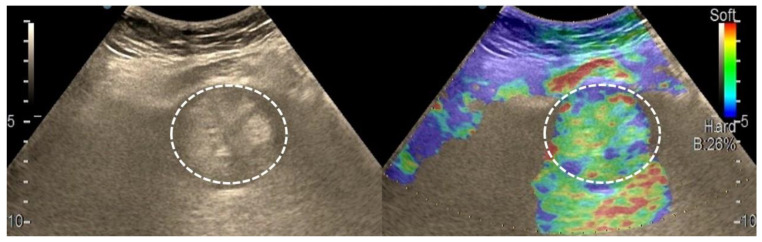
Appearance of hepatocellular carcinoma with type “b” elasticity in a 71-year-old cirrhotic patient (white circles). The lesion has a mosaic pattern with dominant green areas.

**Figure 4 diagnostics-13-03288-f004:**
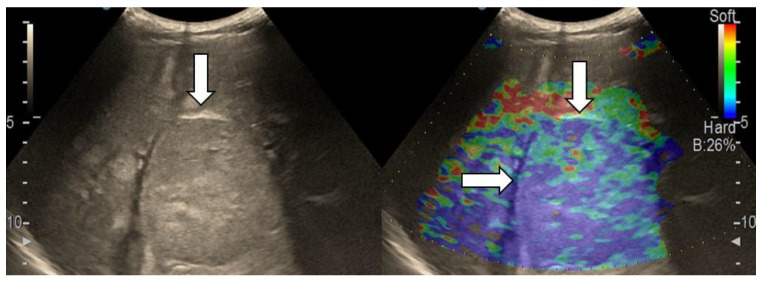
Hepatocellular carcinoma with type “c” elasticity (arrows) in a 61-year-old patient with HCV-related liver cirrhosis. The lesion has a mosaic pattern with dominant blue areas.

**Figure 5 diagnostics-13-03288-f005:**
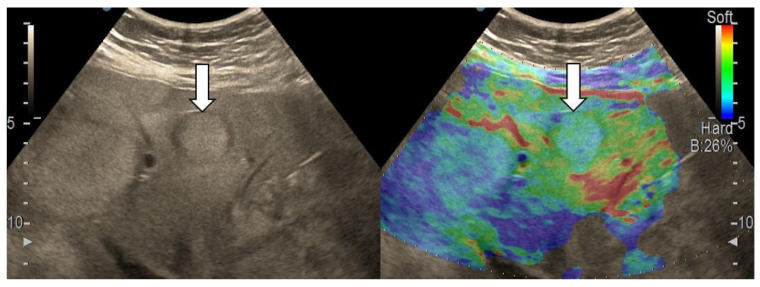
Liver metastasis from small-cell lung carcinoma with type “d” elasticity (arrows) in a 64-year-old patient. The lesion is homogenously light blue.

**Figure 6 diagnostics-13-03288-f006:**
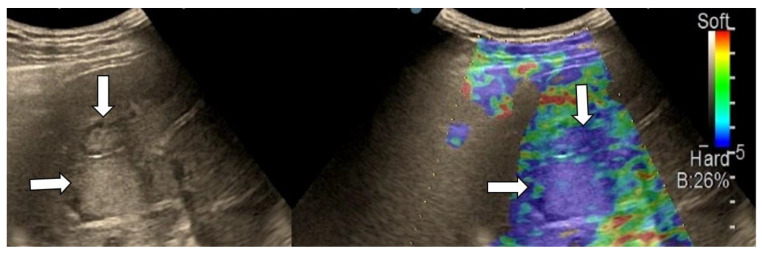
Metastatic liver tumors with type “e” elasticity (white arrows) from colon adenocarcinoma in a 74-year-old patient. The lesions are homogenously deep blue.

**Figure 7 diagnostics-13-03288-f007:**
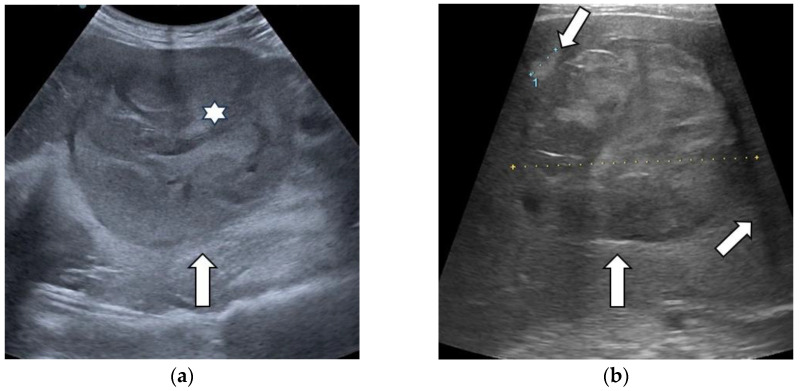
Representative cases of hepatocellular carcinoma (arrow) with “nodule-in-nodule” (asterisk) architecture (**a**), mosaic appearance, satellite nodule and lateral shadow (white arrows) (**b**) in standard ultrasound.

**Figure 8 diagnostics-13-03288-f008:**
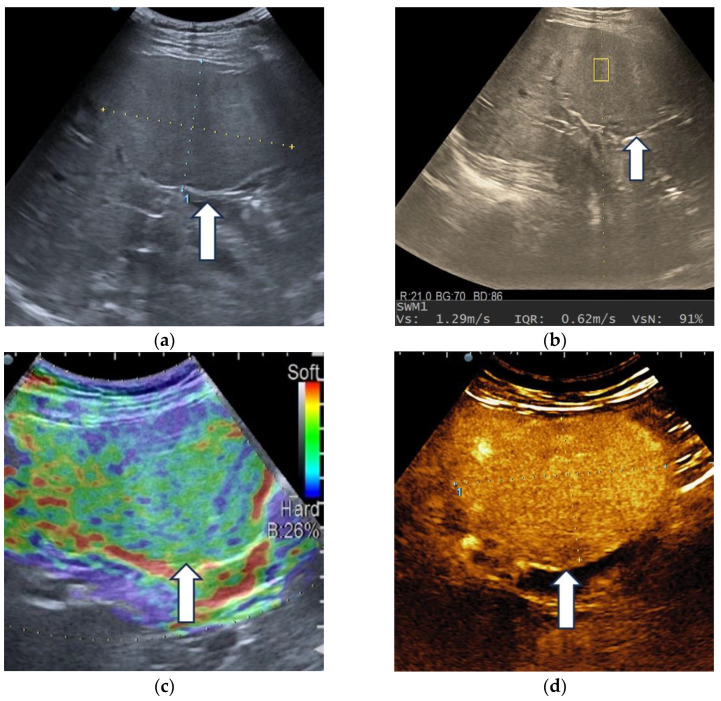
Multiparametric ultrasound evaluation of a liver adenoma in a 60-year-old female patient. B-mode ultrasound showed an isoechoic homogenous focal liver lesion (arrow), with a size of 90/50 mm, located in segment IV (**a**). Shear wave velocity (Vs) obtained in the tumor was 1.29 m/s and the net amount of effective shear wave velocities (VsN) was 91%, suggesting a reliable measurement (**b**). In real-time transabdominal elastography, the lesion showed a mosaic appearance with predominant green areas (arrow), corresponding to type “b” elasticity (**c**). After intravenous administration of the contrast agent, diffuse homogeneous hyperenhancement (arrow) was observed (**d**), followed by sustained enhancement (arrow) during the late phase (**e**), even at 6 min after injection (arrow) (**f**).

**Figure 9 diagnostics-13-03288-f009:**
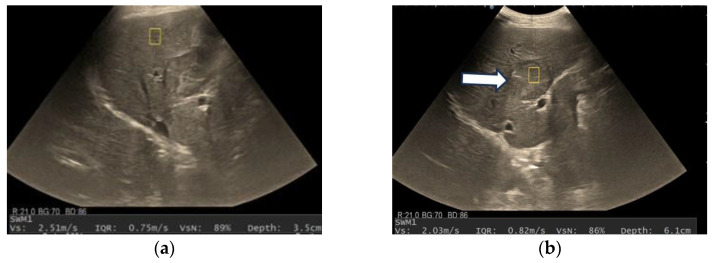
A case of hepatocellular carcinoma developed in a 72-year-old patient with alcoholic cirrhosis. Shear wave velocity in liver parenchyma was 2.51 m/s (19 kPa), which suggests liver fibrosis F4. The measurement was reliable, as indicated by the high VsN (89%) (**a**). Shear wave velocity in the liver tumor was 2.03 m/s, suggesting the lesion is softer than liver parenchyma (**b**). A detailed image of B-mode ultrasound showing a 40 mm, well-delimitated nodule (arrow) in segment IV of the liver (**c**). On contrast-enhanced ultrasound, HCCs showed a diffuse homogenous hyperenhancement (arrow) in the arterial phase (**d**), with sustained enhancement (arrow) in the portal venous phase (**e**) and mild washout (arrow) in the late phase (**f**). In real-time tissue elastography (RTE), the lesion showed a mosaic pattern with predominant blue areas (arrow), i.e., type “c” elasticity (**g**). Pathological examination confirmed the diagnosis of hepatocellular carcinoma (**h**).

**Figure 10 diagnostics-13-03288-f010:**
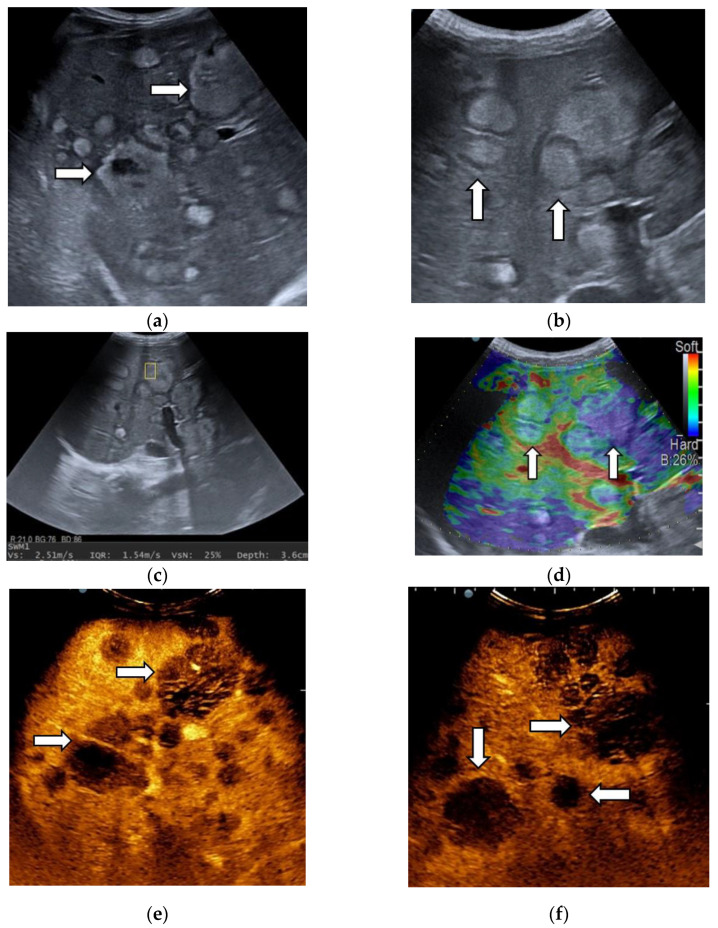
Unquantifiable liver metastases from palatine tonsil squamous cell carcinoma in a 41-year-old patient. In B-mode ultrasound, lesions appear hyperechoic, surrounded by a hypoechoic halo (arrows) and have variable size (**a**,**b**). In point shear-wave elastography using the SWM mode, a shear wave velocity of 2.51 m/s was achieved, suggesting a stiff lesion (**c**), confirmed by real-time tissue elastography, which illustrated the lesions as homogenously deep blue (arrow) (**d**). The other smaller liver metastases were homogenously light blue (arrow) (type “d” elasticity) (**d**). In the arterial phase of CEUS, lesions are hypoenhanced (arrows) (**e**), with marked washout (arrows) in the portal venous phase (**f**).

**Figure 11 diagnostics-13-03288-f011:**
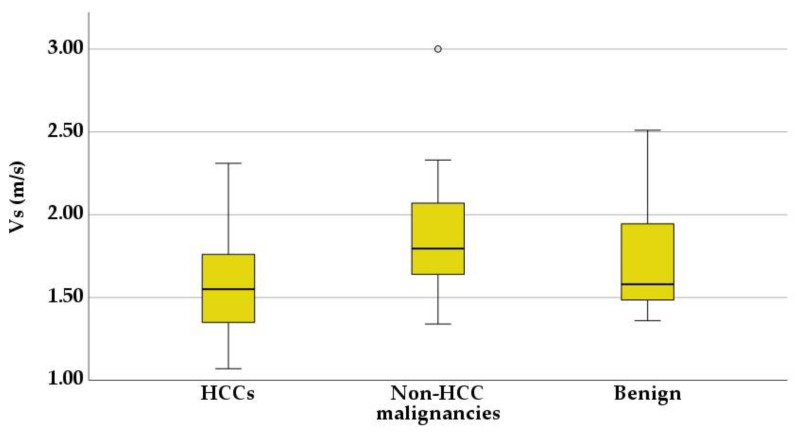
Box-and-whisker plots of shear wave velocity (Vs) in the three tumoral groups.

**Figure 12 diagnostics-13-03288-f012:**
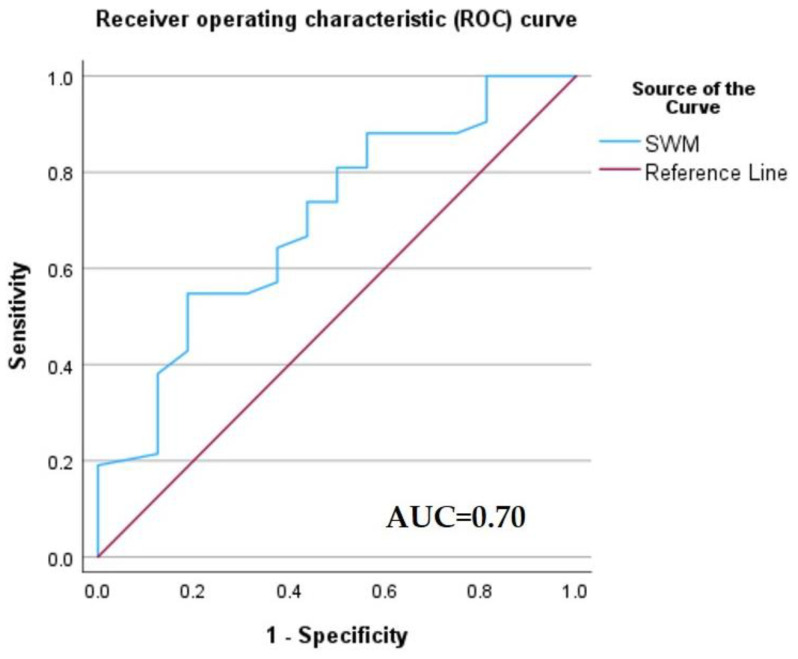
Receiver operating characteristic (ROC) curve of the diagnostic performance of the SWM method for the differentiation of HCCs from non-HCC malignancies.

**Table 1 diagnostics-13-03288-t001:** Classification of elasticity types of focal liver lesions in our study using real-time tissue elastography.

Elasticity Type	Color Code
Type “a”	Homogenously green
Type “b”	Mosaic pattern with dominant green areas
Type “c”	Mosaic pattern with dominant blue areas
Type “d”	Homogenously light blue
Type “e”	Homogenously dark blue

The five categories are based on the elasticity type of liver tumor (ETLT) [23] and modified elasticity type of liver tumors (modETLT) [22] classifications.

**Table 2 diagnostics-13-03288-t002:** The distribution of focal liver tumors according to the final diagnosis.

Final Diagnosis		Number of Lesions
Benign tumors (*n* = 11)	Hepatic adenoma	3
	Hepatic hemangioma	4
	Complicated liver cyst	1
	Gharbi type IV hydatid cyst	1
	Focal nodular hyperplasia	2
Malignant tumors (*n* = 61)	Hepatocellular carcinoma	44
	Non-HCC malignancies	17
	➢ Liver metastasis	10
	➢ Cholangiocarcinoma	6
	➢ Liver lymphoma	1

**Table 3 diagnostics-13-03288-t003:** Clinical characteristics of the patient sample.

Clinical Features	Entire Study Group	HCCs	Non-HCC Malignancies	Benign Tumors
Liver cirrhosis	29 (43.93%)	24 (36.36%)	3 (4.61%)	2 (3.07%)
Chronic hepatitis B and C	17 (26.15%)	13 (19.69%)	2 (3.07%)	2 (3.07%)
Histopathologic evidence of chronic hepatitis (non-HBV, non-HCV)	6 (9.09%)	5 (7.57%)	1 (1.51%)	0
No underlying liver disease	14 (21.21%)	0	9 (13.63%)	5 (7.57%)
History of extrahepatic malignancy	15 (22.72%)	6 (9.09%)	8 (12.12%)	1 (1.51%)
Ascites	14 (21.21%)	12 (18.18%)	2 (3.07%)	0
Portal vein thrombosis	12 (18.18%)	10 (15.15%)	2 (3.07%)	0

HBV: hepatitis B virus; HCV: hepatitis C virus; HCC: hepatocellular carcinoma.

**Table 4 diagnostics-13-03288-t004:** B-mode ultrasound features of liver tumors.

Ultrasound Features		Entire Group	HCCs	Non-HCC Malignancies	Benign Tumors
Size (mm)		70.11 ± 38.97	76.73 ± 40.8	60.88 ± 30.86	57.93 ± 35.49
Echogenicity	Hypoechoic	22 (30.6%)	12 (16.67%)	7 (9.72%)	3 (4.17%)
	Isoechoic	14 (19.4%)	9 (12.50%)	2 (2.78%)	3 (4.17%)
	Hyperechoic	31 (43.1%)	21 (29.17%)	5 (6.94%)	5 (6.94%)
	Mixed echogenicity	5 (6.9%)	2 (2.78%)	3 (4.17%)	0
Homogeneity	Homogenous	5 (6.9%)	2 (2.78%)	1 (1.39%)	2 (2.78%)
	Non-homogenous	67 (93.1%)	42 (58.33%)	16 (22.22%)	9 (12.50%)
Tumor boundaries	Well-delimited	58 (80.6%)	38 (52.78%)	12 (16.67%)	8 (11.11%)
	Ill-delimited	14 (19.4%)	6 (8.33%)	5 (6.94%)	3 (4.17%)
"Nodule-in-nodule” architecture	Yes	18 (25%)	17 (23.61%)	1 (1.39%)	0
	No	54 (75%)	27 (37.50%)	16 (22.22%)	11 (15.28%)
Mosaic appearance	Yes	16 (22.2%)	16 (22.2%)	0	0
	No	56 (77.8%)	28 (38.89%)	17 (23.61%)	11 (15.28%)
Halo sign	Yes	28 (38.9%)	21 (29.17%)	5 (6.94%)	2 (2.78%)
	No	44 (61.1%)	23 (31.94%)	12 (16.67%)	9 (12.50%)
Satellite nodules	Yes	8 (11.1%)	7 (9.72%)	1 (1.39%)	0
	No	64 (88.9%)	37 (51.39%)	16 (22.22%)	11 (15.28%)

HCC: hepatocellular carcinoma.

**Table 5 diagnostics-13-03288-t005:** CEUS performance for hepatocellular carcinoma diagnosis.

Sensitivity (95% CI)	Specificity (95% CI)	PPV (95% CI)	NPV (95% CI)	Accuracy (95% CI)
69.05%	92.86%	93.55%	66.67%	78.57%
(52.91–82.39%)	(76.50–99.12%)	(78.97–98.24%)	(55.72–76.07%)	(67.13–87.48%)

95% CI: 95% confidence interval; PPV: positive predictive value; NPV: negative predictive value.

**Table 6 diagnostics-13-03288-t006:** Shear wave velocities for liver tumors.

Tumor Type	Mean Vs (m/s)	Range
HCCs	1.59 ± 0.29	1.07–2.31 m/s
Non-HCC malignancies	1.9 ± 0.42	1.34–3 m/s
Benign tumors	1.75 ± 0.4	1.36–2.51 m/s

Vs: Shear wave velocity; HCC: hepatocellular carcinoma.

**Table 7 diagnostics-13-03288-t007:** SWM performance for hepatocellular carcinoma diagnosis.

Sensitivity (95% CI)	Specificity (95% CI)	PPV (95% CI)	NPV (95% CI)	Accuracy (95% CI)
54.76%	82.35%	88.46%	42.42%	62.71%
(38.67–70.15%)	(56.57–96.20%)	(72.59–95.69%)	(33.09–52.34%)	(49.15–74.96%)

95% CI: 95% confidence interval; PPV: positive predictive value; NPV: negative predictive value.

**Table 8 diagnostics-13-03288-t008:** Characteristics of the two VsN groups.

Variables	VsN ≥ 50%	VsN < 50%	*p*-Value
Vs (m/s)	1.66 ± 0.3	1.73 ± 0.44	0.74
Depth of the lesion (cm)	5.14 ± 1.16	5.73 ± 1.60	1.31
Size (mm)	75.59 ± 29.62	66.17 ± 45	0.06
Presence of liver cirrhosis	17	15	0.34

Vs: shear wave velocity; VsN: net amount of effective shear wave velocity (%).

**Table 9 diagnostics-13-03288-t009:** Distribution of focal liver lesions according to the elasticity model.

RTE	Liver Adenoma	HMG	FNH	Complicated Liver Cyst	HCC	iCCA	Liver Metastases	Liver Lymphoma
Type “a”		3		1				
Type “b”	1	1			10		3	1
Type “c”			1		24	3	6	
Type “d”							1	
Type “e”					5	3	1	

RTE: real-time tissue elastography; HMG: hemangioma; FNH: focal nodular hyperplasia; HCC: hepatocellular carcinoma; iCCA: intrahepatic cholangiocarcinoma.

**Table 10 diagnostics-13-03288-t010:** Diagnostic performance of combined SWM and CEUS in differentiating HCCs from other groups.

Sensitivity (95% CI)	Specificity (95% CI)	PPV (95% CI)	NPV (95% CI)	Accuracy (95% CI)
79.55%	89.29%	92.11%	73.53%	83.33%
(64.70–90.20%)	(71.77–97.73%)	(79.85–97.17%)	(60.47–83.46%)	(72.70–91.08%)

95% CI: 95% confidence interval; PPV: positive predictive value; NPV: negative predictive value.

## Data Availability

The data used to support the findings of this study are available from the corresponding author upon request.

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
