# Peer review of "The Clinical Value of Multimodal Ultrasound for the Differential Diagnosis of Hepatocellular Carcinoma from Other Liver Tumors in Relation to Histopathology"

_diagnostics, 2023, doi:10.3390/diagnostics13203288_

Round 1

Reviewer 1 Report

The article is a comprehensive study on focal liver lesions, employing a multi-modal ultrasound technique for diagnosis. It is a laudable attempt to offer a clinical framework for assessing liver lesions and distinguishing between benign and malignant types.

Given the rising incidence of liver lesions and the necessity for a precise diagnostic protocol, the topic is clinically relevant.

It will be interesting to analyze if SWE or real-time elastography over CEUS will increase the performances of diagnostics. That would be a true multiparametric analysis.

Minor language polishing is needed. 

Reviewer 2 Report

I read with great interest the manuscript :The clinical value of multimodal ultrasound for the differential diagnosis of hepatocellular carcinoma from other liver tumors  in relation to histopathology

would like to congratulate with the authors. The paper is well presented with adequate images and correct statistical analysis. 

Please on line 48 "The value of CEUS in differentiating malignant and benign tumors has been well established." I would add a reference to this sentence authors may rely on this manuscript. 

doi: 10.3390/diagnostics12051209.

Other minors: Typos and grammar check.
